# An Automatic UAV Based Segmentation Approach for Pruning Biomass Estimation in Irregularly Spaced Chestnut Orchards

**Salvatore Filippo Di Gennaro** [1], **Carla Nati** [1,*], **Riccardo Dainelli** [1], **Laura Pastonchi** [1], **Andrea Berton** [2], **Piero Toscano** [1] and **Alessandro Matese** [1]

[1] Institute of BioEconomy (IBE), National Research Council (CNR), Via Caproni 8, 50145 Florence, Italy; salvatorefilippo.digennaro@cnr.it (S.F.D.G.); riccardo.dainelli@ibe.cnr.it (R.D.); laura.pastonchi@ibe.cnr.it (L.P.); piero.toscano@cnr.it (P.T.); alessandro.matese@cnr.it (A.M.)

[2] Institute of Clinical Physiology (IFC), National Research Council (CNR), Via Moruzzi 1, 56124 Pisa, Italy; bertonandrea@ifc.cnr.it

* Correspondence: carla.nati@ibe.cnr.it; Tel.: +39-055-5225640

**Abstract:** The agricultural and forestry sector is constantly evolving, also through the increased use of precision technologies including Remote Sensing (RS). Remotely biomass estimation (WaSfM) in wood production forests is already debated in the literature, but there is a lack of knowledge in quantifying pruning residues from canopy management. The aim of the present study was to verify the reliability of RS techniques for the estimation of pruning biomass through differences in the volume of canopy trees and to evaluate the performance of an unsupervised segmentation methodology as a feasible tool for the analysis of large areas. Remote sensed data were acquired on four uneven-aged and irregularly spaced chestnut orchards in Central Italy by an Unmanned Aerial Vehicle (UAV) equipped with a multispectral camera. Chestnut geometric features were extracted using both supervised and unsupervised crown segmentation and then applying a double filtering process based on Canopy Height Model (CHM) and vegetation index threshold. The results show that UAV monitoring provides good performance in detecting biomass reduction after pruning, despite some differences between the trees' geometric features. The proposed unsupervised methodology for tree detection and vegetation cover evaluation purposes showed good performance, with a low undetected tree percentage value (1.7%). Comparing crown projected volume reduction extracted by means of supervised and unsupervised approach, $R^2$ ranged from 0.76 to 0.95 among all the sites. Finally, the validation step was assessed by evaluating correlations between measured and estimated pruning wood biomass (Wpw) for single and grouped sites ($0.53 < R^2 < 0.83$). The method described in this work could provide effective strategic support for chestnut orchard management in line with a precision agriculture approach. In the context of the Circular Economy, a fast and cost-effective tool able to estimate the amounts of wastes available as by-products such as chestnut pruning residues can be included in an alternative and virtuous supply chain.

**Keywords:** unmanned aerial vehicles; precision agriculture; biomass evaluation; image processing; *Castanea sativa*

## 1. Introduction

Remote Sensing (RS) is one of the technologies that has been currently most employed in the forestry sector for monitoring, inventorying, and mapping purposes. RS techniques with the aim to obtain information on large areas can be conducted at different levels of precision, according to the different goals to be achieved. The choice of the RS platform to be employed, and consequently the

sensors installed and operating on-board that specific platform will depend on the processes under investigation and the level of detail required for a particular analysis [1].

RS platform as satellite systems, aircraft platforms and unmanned aerial vehicles (UAVs) have features that differ in terms of spatial resolution, surface covered, temporal resolution, operational procedures, and costs. Satellite solutions remain a fundamental tool for long-term and extensive monitoring and surveillance forestry activities against fire events [2], pests attack [3], illegal logging [4] and more generally, to assess the health and structure of forests' cover [5]. Aircraft platforms provide a better image resolution, returning a higher level of detail compared to satellite, against a higher effort in flight planning and relevant operational costs [6]. UAVs are flexible small platforms characterized by low operational costs, high spatial and temporal resolution [7] but suitable to cover only limited areas. Comparisons among different platforms have been made both in the agricultural [8] and in the forestry field [9].

The use of UAV in precision forestry has exponentially increased in recent years, as demonstrated by the large number of papers published between 2018 and 2019; more than 400 references were found when searching for "UAV" + "forest" and considering articles, conference proceedings and books [10].

Authors have dealt with several research topics involving applications in forest monitoring, inventorying, and mapping both with multirotor and fixed-wing unmanned platforms equipped with a wide series of optical technology sensors [11–18]. These studies took into account forestry UAV applications mainly within two forest types: the first one included planted, pure and even-aged forests [19–24] and the second one included natural, mixed and uneven-aged forests where the spatial variability of vegetation was very high [25–29].

Within natural, mixed and uneven-aged forests research, UAVs have been employed most commonly for (i) estimation of dendrometric parameters such as dominant height, stem number, crown area, volume and above-ground biomass (Wa) using RGB (Red–Green–Blue bands camera) [30–34], multispectral near red green (NRG) [35,36] and laser scanning [37,38] sensors. This is the top research topic because reliable information on the status and trends of forest resources is the basis for the decision-making process for forest management and planning [39]; (ii) tree species classification and invasive plants detection for forest inventories and monitoring of biodiversity using RGB [40,41], multispectral [42,43], hyperspectral [29,44] and laser scanning [45] sensors; (iii) flight plan ad RGB sensor settings to improve imagery products accuracy [26,46–49] (iv) forest health monitoring and diseases mapping using different sensors (RGB [50], multispectral [51], hyperspectral [52], thermal [15]) to provide data for supporting intervention decisions in the management of forests; (v) recovery monitoring after fire events or conservation interventions through UAV equipped with RGB [53] and multispectral [2,17] cameras.

By providing key forest structural attributes such as tree crown centers and boundaries, UAV imagery tree segmentation is used for stem counting [32,54], extrapolation of further dendrometric parameters (i.e., Wa) [55–59], species recognition [42], and pathogens detection and mapping [60].

Regarding Wa estimation, there are two main strategies adopted for Digital Aerial Photogrammetry (DAP) and Airborne Laser Scanning (ALS)-based analysis in forestry inventories: (i) the Area-Based Approach (ABA), a distribution-based technique which provides data at stand level using predictive models developed with co-located ground plot measurements and RS data that are then applied to the entire area of interest to generate estimates of specific forest attributes [61]; and (ii) Individual Tree Crown segmentation (ITC) delineation, in which individual tree crowns, heights and positions are the basic units of assessment and where specific algorithms are used to identify the location and size of individual trees from raster images or high-density point clouds [62]. Previous research papers dealt with biomass estimation both at the stand and at tree level. Biomass at stand level is evaluated by comparing the effects of flight settings, sensor type and resolution in tropical woodlands [55], the influence of plot size in dry tropical forests [58] or by taking into account different mangrove species in South China wetlands [59]. Concerning tree-level biomass estimation, Guerra-Hernandez et al. [57] and Guerra-Hernandez et al. [56] used, respectively, UAV-DAP point clouds in open Mediterranean

forest of coniferous *Pinus pinea* (Central of Portugal) and DAP and ALS data in evergreen *Eucalyptus spp.* plantation (North of Portugal). The latter two studies are important references for modeling SfM individual tree diameters and SfM-derived individual tree biomass ($Wa_{SfM}$) and volume ($V_{SfM}$) from the canopy height model (CHM) in Mediterranean forest plantation.

Segmentation of individual tree crowns is difficult, particularly in broadleaf, mixed, or multi-layered forests. This is generally due to an inability to determine the appropriate kernel size to simultaneously minimize omission and commission error with respect to tree stem identification [63]. In the literature, several unsupervised segmentation approaches have been proposed: the most widely used is the watershed segmentation algorithm [20,23,34,64–67] and its variants [16,54,68]. Other techniques are multiresolution segmentation algorithm [27,69], large-scale mean-shift algorithm [35], semantic-level segmentation using a Convolutional Neural Network (CNN) [70] and more complex approaches with two or more integrated algorithms [37,63,71]. Some authors associated the above-mentioned unsupervised approaches to manually drawn individual tree crown polygons from on-screen interpretation to compare and validate results or provide a reference for the accuracy assessment of an automatic procedure [72–74].

Among the papers that adopted both manual and unsupervised tree segmentation, only a few research works included ground data collection [42,72,75–77] with a tree sample size ranging from 109 to 2069 trees. None of those presented wood biomass in-field data. For natural, mixed and uneven-aged forest, Mayr et al. [75] gathered tree height in dry savannah and used an implementation of the watershed segmentation algorithm provided by System for Automated Geoscientific Analyses-Geographic Information System (SAGA-GIS) while Franklin and Ahmed [42] utilized the multi-resolution segmentation procedure with the ENVI software system and they collected tree height and crown dimensions in a mixed maple, aspen, and birch forest. Concerning planted, pure and even-aged forests, Ganz et al. [72] used a multiresolution segmentation algorithm and measured tree height within stands of Norway spruce and common beech while Apostol et al. [77] utilized the watershed algorithm and collected tree height and stem diameter in an even-aged Douglas fir stand. By taking tree height as ground-truth data in a chestnut plantation, Marques et al. [76] segmented trees by combining a vegetation-index based algorithm with the Otsu method.

Chestnut (*Castanea sativa Mill.*) orchards are a type of multifunctional tree cultivation used worldwide that represent a relevant income for rural populations. In Italy, sweet chestnut groves cover 147,568 hectares (ha) of the whole Italian forested territory [78]. Only a few research papers used UAV in chestnut plantations and dealt with phytosanitary problem detection and monitoring of tree health [76,79,80], automatic classification and segmentation of chestnut fruits through Convolutional Neural Networks (CNNs) [81], and insects damage rate detection and pest control methods [82]. However, there is no research available that tried to estimate the amounts of residues coming from tree tending by using UAV techniques and comparing their information with ground truth. In the present study, the authors applied RS techniques (UAV) to collect data on uneven-aged and irregularly spaced chestnut (*Castanea sativa Mill.*) orchards. The aim of the present study was to verify the reliability of RS techniques for the estimation of pruning wood biomass (Wpw) through differences in the volume of canopy trees calculated with supervised extraction and to evaluate the performance of an unsupervised segmentation methodology as a feasible tool for large-area analysis. In the context of the Circular Economy, a fast and cost-effective tool able to estimate the amounts of residues available as by-products, such as chestnut pruning material, can be included in an alternative and virtuous supply chain.

## 2. Materials and Methods

### 2.1. Experimental Sites

The study took place within the Amiata mountain region (Tuscany, Italy) between 2017 and 2018. Four sites located into three different chestnut orchards were selected as representative of this area in terms of variety and management practices (Figure 1).

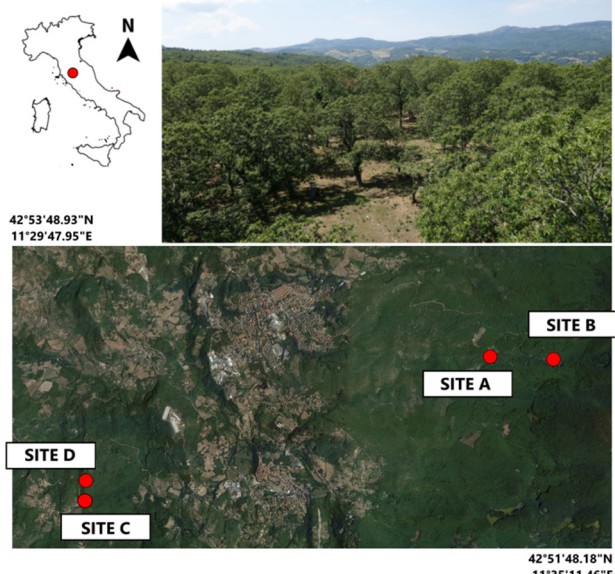

**Figure 1.** Experimental sites map.

The experimental sites' characteristics are shown in Table 1. The chestnut orchards under study were uneven-aged and irregularly spaced (Figure 1), mainly due to replacements of dead trees that occurred over time. For this reason, they were equated to irregular forests.

**Table 1.** Experimental sites' description.

| ID | Site A | Site B | Site C | Site D |
|---|---|---|---|---|
| Location | 42°53'18.22" N 11°33'41.57" E | 42°53'17.19" N 11°33'41.75" E | 42°52'11.71" N 11°30'28.55" E | 42°52'18.59" N 11°30'29.65" E |
| Altitude (m ASL) | 960 | 1085 | 780 | 755 |
| Surface (ha) | 0.55 | 0.32 | 0.36 | 0.32 |
| Chestnut variety | Cecio | Cecio | Bastarda Rossa | Bastarda Rossa |
| Density (trees ha$^{-1}$) | 72.57 | 114.60 | 110.10 | 111.73 |
| Canopy cover (%) | 82.50 | 86.47 | 90.19 | 87.80 |

*2.2. Pruning Wood Biomass Ground Measurement*

At the beginning of March 2017, 30 chestnut trees per plot (A, B, C, D) were selected and the diameter at breast height (DBH) was callipered. The choice of the sample trees was made by identifying plants representative of each site in terms of size (DBH). Sample trees were georeferenced at high resolution (0.02 m) with a differential GPS (Leica GS09 GNSS, Leica Geosystems AG). In February 2018, the previously selected trees were pruned, their branches severed and grouped in two sets: "wings" (pieces below 4 cm in diameter) and "wood" (pieces over 4 cm in diameter). The first group had no commercial use while the second could follow two different destinations: sold as firewood after being seasoned in the field or sold to the industry for tannins extraction. These two raw materials were separately loaded on a tractor equipped with a bucket or a pitchfork and weighed by means of portable scales (model WWSD6T, Nonis s.r.l., Biella, Italy). Every five weighing the scales' accuracy was checked by weighing the tractor unloaded. From every site, Wpw samples were collected and weighed fresh, then oven-dried according to the standard UNI EN ISO 18134-2:2017 to measure their moisture content. The following analyses and comparisons were made on a dry matter basis, avoiding uncontrollable variability due to wood samples size, initial moisture conditions or seasoning.

*2.3. UAV Platform and Data Processing*

Remote sensed data were acquired to characterize the intra-plot variability in terms of plant vigor and Wpw. Two flight campaigns were performed on 2 August 2017 and on 25 July 2018 at the same phenological stage using a modified multi-rotor Mikrokopter (HiSystems GmbH, Moomerland, Germany) described in Matese and Di Gennaro [83] equipped with a multispectral camera Tetracam ADC Snap (Tetracam Inc., Chatsworth, CA, USA). The second flight was performed immediately after the canopy pruning management to enable a comparison between ground truth and UAV results. Multispectral image acquisition was planned flying at 60 m above ground level at midday, yielding a ground resolution of 0.05 m pixel$^{-1}$ and a 70% overlap in both directions. The images were recorded in clear sky conditions. The radiometric calibration processes were realized by acquiring, during the flight, images from three OptoPolymer (OptoPolymer-Werner Sanftenberg, Munich, Germany) reference panels, with 95%, 50%, and 5% reflectance, respectively.

The data processing workflow is described in Figure 2. Multispectral or NRG images with three broad bands (Near-infrared–Red–Green bands) acquired by UAV were processed using Agisoft Metashape Professional Edition 1.5.2 [84], which allows to generate the dense cloud and the orthomosaic of each experimental site. During this process, any ground control points (GCPs) were used due to the irregular and dense canopy cover. The spatial variability in the chestnut orchard was evaluated in terms of vigor and assuming the correspondence between NDVI and vigor [85,86]. NDVI was used as a further filter threshold, as described in Section 2.4.

The dense cloud obtained was normalized using a digital elevation model (DEM) from the automatic classification of ground points from photogrammetric software and subsequently imported into QGis software [87] to develop, by means of the LAStools toolbox [88], the CHM relative to the canopy height of each sample tree. The resolution chosen for this model was 0.05 m.

The next processing step concerned the creation of a chestnut crown mask through a two-fold approach: supervised and unsupervised segmentation. The supervised method consisted of manually drawing each chestnut crown one by one within the experimental plot, visualizing together the CHM and the NRG orthomosaic in the QGis software. The unsupervised approach used a script called 'rLIDAR' (version 0.1.1) [89] in R programming language (version 3.6.0), which allows to generate a vector format file relative to the position and the crown dimension of each sample tree. First, CHM smoothing was performed to eliminate spurious local maxima caused by tree branches. Then, the location and height of individual trees were automatically detected using the CHM and the Local maxima method (rLiDAR: FindTreesCHM function) by sequentially searching the moving window through a Fixed Window Size (FWS) set to 9x9 pixels. In this step, we used a lower CHM resolution (0.25 m/pixel) to generate the mask with the unsupervised method, due to the fact that the workflow with native resolution (0.05 m/pixel) required to many computing resources. However, the segmentation provided enough accuracy with respect to the supervised segmentation. The threshold for the lowest tree height (minht) was fixed at 3.0 m to avoid the misdetection of forest undergrowth as trees. For unsupervised crown segmentation, the ForestCAS function (cf. rLiDAR) based on the watershed method was applied to automatically detect crown boundaries. The threshold for the maximum crown radius (maxcrown) was set to 15.0 m, according to chestnut dendrometric characteristics.

Finally, the obtained dataset was analyzed to perform a spatial estimation of the potential pruning biomass (Figure 2). The tree crown volume was calculated at the pixel level by integrating the volume of all the individual pixels that were positioned below each tree. This choice was made to deal with the irregular shape of every tree and consequently, to reduce the error usually produced in empirical estimations due to the inexact assimilation of trees to regular solids. Therefore, as suggested by Torres-Sanchez et al. [90], the height and area of every tree pixel were multiplied to obtain the pixel volume; subsequently, the crown projected volume was derived by adding the volume of all the pixels below each chestnut tree.

For identifying the volume change between the two years (before and after pruning), the tree mask generated both in the supervised and unsupervised method for the 2017 dataset was chosen

as the reference and also used in 2018 (QGis Software) to correct the XY shift of the camera GPS between the two flights. This operation avoided an overestimation of the segmented crown area in 2018 following pruning, especially with R software. The Wpw estimation was performed on the basis of the crown volume reduction in the post-pruning survey with respect to the first flight. A linear regression model between the Wpw measured on the ground and estimated by UAV was applied to evaluate the performance of the UAV approach.

Since the aim of this work was Wpw estimation from remote sensing data, for ground truth measurements, we focused on a very large number of pruning wood sampling (30 trees/site), while only the DBH parameter was meausured as geometric field data for sample trees selection. As a consequence, the geometric data evaluation was related only to the comparison between the supervised and unsupervised methods on the structure from the motion dataset, without any field data as ground truth (tree height or crown dimension).

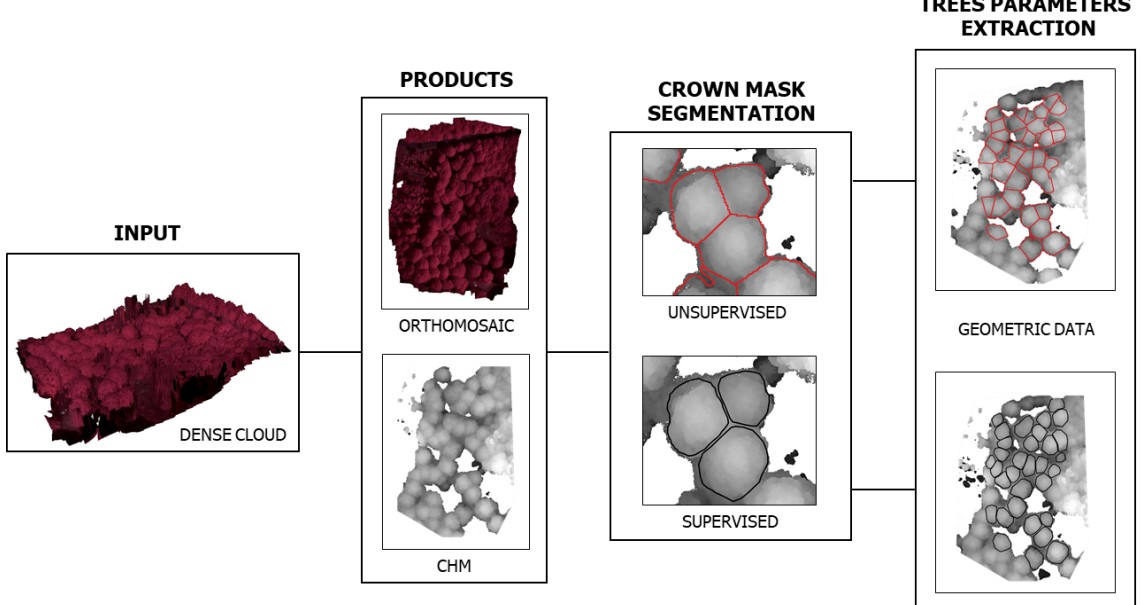

**Figure 2.** UAV elaboration data workflow.

### 2.4. Double Filtering Approach

Several authors reported on the improvement in tree crown segmentation when vegetation indices analysis is applied in discriminating between vegetation targets [76,90]. However, in our study, the discrimination between canopy and no canopy pixels was ensured by the CHM thanks to the higher tree height which was more than double the regular plantation sites observed in other works [76,90]. Although the spectral data were not used to improve the segmentation of the crowns from the soil, in our study, they were used to improve the measurement of volume reduction from the CHM.

In detail, the elaboration of crown volume data from the pruned tree (2018 survey) accounted also for no canopy information of the small holes within the canopy undetected by the 3D reconstruction process performed with Agisoft Metashape but clearly visible in the orthomosaic. To solve these problems, we applied a double filtering process: the first one based on canopy height (CHM) and the second based on a vegetation index (NDVI) threshold to remove no vegetation pixels within the crown (Figure 3). The results presented in this work were obtained from a dataset filtered with an NDVI threshold of 0.3.

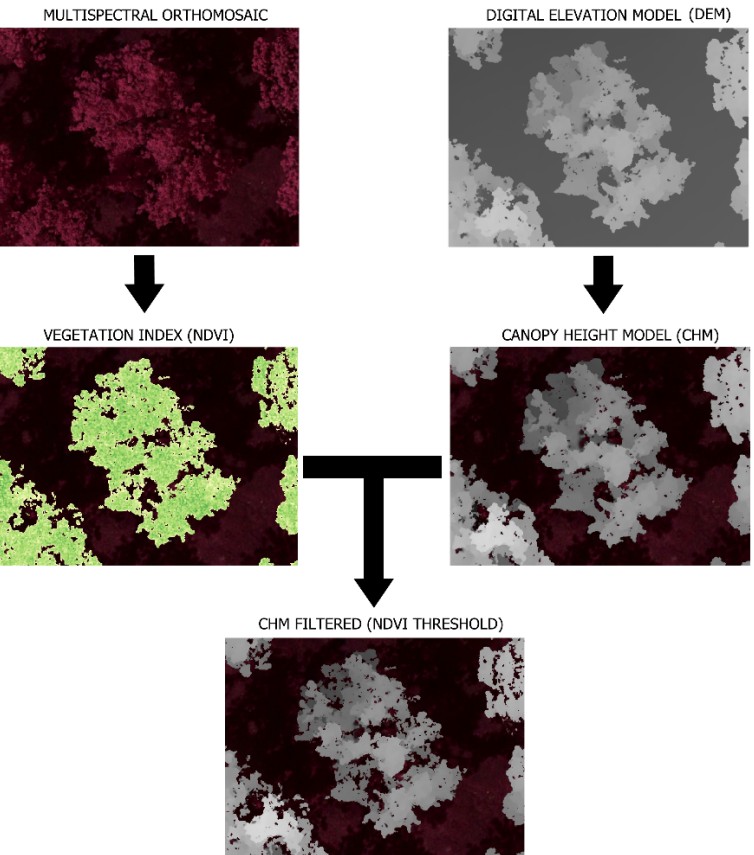

**Figure 3.** Filtering workflow aimed to improve the accuracy of the canopy height model.

*2.5. Puning Wood Biomass Estimation*

Reference segmentation masks of sample trees were manually created for each experimental site. The supervised method was applied to develop a linear model between ground truth Wpw and crown projected volume reduction extracted by the reference segmentation mask. The linear model was then applied to calculate the estimated Wpw from the unsupervised segmentation method following Equation (1):

$$Y = \beta (X_1 - X_2) + \gamma \tag{1}$$

where Y is the dependent variable (Wpw), $X_1$ to $X_2$ are independent variables related, respectively, to crown-projected volume before and after pruning management, β is the multiplicative parameter and γ is the intercept. The coefficient of determination ($R^2$) and Root Mean Square Error (RMSE) were computed between the measured correlations, between supervised and unsupervised segmentation approaches for UAV geometric data extraction, and between ground-truth measured and estimated Wpw data.

The adjusted coefficient of determination (Equation (2)), the relative root mean square error (Equation (3)) and the percentage bias (Equation (4)) to determine the accuracy of unsupervised segmentation for estimating Wpw using crown projected volume reduction are as follows:

$$adjR2 = 1 - ((n-1)\sum(yini = 1 - \hat{y}i)2(n-p)\sum(yini = 1 - \overline{y}i)2) \tag{2}$$

$$\text{rRMSE} = RMSE\overline{y} \tag{3}$$

$$\text{PBias} = 100*(\sum(\hat{y}i - yini = 1)n) \tag{4}$$

where n is the number of trees, yi is the field measured Wpw i, $\overline{y}$ is the the mean observed value of Wpw and $\hat{y}i$ is the estimated value of Wpw derived from the linear regression model. A statistical analysis was performed using R software.

## 3. Results

### 3.1. Wpw Ground Measurement

In Table 2 are presented the ground data of measured Wpw per tree, per site (sample of 30 trees) and per surface unit.

**Table 2.** Pruning wood biomass (Wpw) produced per tree (Avg. and Std. dev.), per plot, and per surface.

| SiteTitle | DBH (cm) | Wpw per Tree ($kg_{dw}$) | Wpw per Site ($kg_{dw}$) | Wpw per Surface ($Mg_{dw}$ $ha^{-1}$) |
|---|---|---|---|---|
| A | 84.21 ± 26.55 | 625.01 ± 590.48 | 18750.28 | 24.92 |
| B | 63.24 ± 11.97 | 243.69 ± 90.42 | 7310.85 | 8.94 |
| C | 49.77 ± 11.75 | 97.37 ± 63.99 | 3115.84 | 3.86 |
| D | 53.38 ± 15.5 | 113 ± 76.39 | 3390.00 | 4.05 |

Note: dw = dry weight.

The yield of Wpw per hectare was comparable with the data presented in a former study [91], where three chestnut groves produced from 22 up to 33 Mg $ha^{-1}$, but this result matches only with site A. In fact, although site A was different from the other three sites presented in the aforementioned study, showing an even bigger DBH on average compared to them, it had a similar pruning intensity. Compared to site A, the number of pruning residues produced in the other three sites investigated in the present study turned out to be noticeably lower, probably caused here—as in other case studies—by differences in trees age, site density, and pruning intensity. The results of sites C and D were very similar and still comparable to site B in terms of wood biomass recovered after pruning, despite the different chestnut varieties.

The proportion of "wood" compared to "wings" was equal to 311.9% at site A, 55.7% at site B and 51.7% at site D. At site C, it was not possible to separate the two fractions due to operative reasons.

### 3.2. Supervised Data Extraction

Table 3 shows the geometric characterization of each experimental site arising from supervised segmentation before (2017) and after pruning (2018). Maximum tree height, crown mean height, crown area, and crown projected volume are mean values of each site. Crown area and crown projected volume per site are also showed to provide a general overview of biomass reduction after pruning. These values derive from a single tree crown area and projected crown volume, respectively, multiplied by the total number of trees for each site.

The decrease in heights after pruning is not significantly different. However, considering crown mean height and crown area, biomass reduction between the two years detected by UAV is relevant. In fact, it ranges from 8.8% in site A to 14.2% in site B, referring to the crown mean height and from 6.5% in site B to 15.1% in site A, considering the crown area values. In sites C and D, these two parameters show intermediate but comparable variations, reflecting the geomorphological and vegetational similarities of the two sites, in detail: crown mean heights of 11.0% (C) and 11.6% (D), and crown areas of 13.1% (C) and 11.8% (D).

The tree geometric characteristic that best shows the effects of pruning is crown projected volume, whose values have the strongest variations between 2017 and 2018. The highest percentage of biomass reduction was found at site C (21.4%) while site A has the maximum decrease (298 $m^3$), confirming ground measurements (see Table 2).

**Table 3.** Geometric characterization of each experimental site.

| Year | Site | Tree Height (m) | Crown Mean Height (m) | Crown Area (m$^2$) | Crown Projected Volume (m$^3$) | Crown Area per Site (m$^2$) | Crown Projected Volume per Site (m$^3$) |
|---|---|---|---|---|---|---|---|
| 2017 | A | 18.07 ± 2.8 | 14.69 ± 2.67 | 93.97 ± 49.56 | 1401.66 ± 807.53 | 3006.89 | 44,853.18 |
| | B | 15.19 ± 1.38 | 12.90 ± 1.09 | 59.44 ± 20.32 | 767.61 ± 267.9 | 1902.10 | 24,563.55 |
| | C | 9.63 ± 0.98 | 7.65 ± 0.94 | 61.19 ± 18.69 | 472.7 ± 170.34 | 1958.17 | 15,126.38 |
| | D | 11.23 ± 1.36 | 8.91 ± 1.15 | 63.06 ± 18.70 | 575.36 ± 216.96 | 1954.86 | 17,836.20 |
| 2018 | A | 17.63 ± 2.66 | 13.40 ± 2.58 | 79.74 ± 45.82 | 1103.71 ± 751.49 | 2551.80 | 35,318.81 |
| | B | 14.63 ± 1.36 | 11.07 ± 1.56 | 55.57 ± 20.56 | 620.82 ± 248.39 | 1778.20 | 19,866.23 |
| | C | 9.44 ± 1.12 | 6.81 ± 1.04 | 53.16 ± 19.55 | 371.31 ± 164.53 | 1700.97 | 11,882.00 |
| | D | 11.08 ± 1.68 | 7.88 ± 1.33 | 55.62 ± 22.97 | 460.28 ± 226.76 | 1724.20 | 14,268.58 |

*3.3. Unsupervised Data Extraction*

Table 4 reports the segmentation results of the proposed unsupervised methodology. Following Marques et al. [76]'s study, the evaluation of the automatic segmentation accuracy applied in this work was assessed by comparing it with a manual crowns' segmentation. In line with the different site conditions in terms of trees age and dimension, the proposed methodology provides a different response in terms of accuracy. Site A presented the lowest accuracy value (46.7%) due to the highest presence of both over and under detection cases. In detail, the irregular and oversize crown (mean values over 90 m$^2$) caused elevated crown shape fragmentation (33.3%), while the high overlap crown level led to 20.0% of merged cases. Sites B and D, characterized by a lower overlap level, showed the best accuracy performances, respectively 83.3% and 76.7%. Site D presented a lower accuracy value due to the 20.0% of merged crowns in a circumscribed zone with close trees with similar crown heights. An intermediate accuracy performance was found in the C site (63.3%), where the lowest values and variability in terms of height and some irregular shape cases caused an overestimation of 26.7% crown shape segmentation. The methodology provided the optimal results in terms of undetected tree crown, with 1.7% mean accuracy considering the overall dataset (four sites).

**Table 4.** Report of the trees' detection accuracy with the number of estimated trees and its detection type in the four sites.

| Site | Reference Crowns | Matched | Split | Merged | Missed |
|---|---|---|---|---|---|
| A | 30 | 46.7% | 33.3% | 20.0% | 0.0% |
| B | 30 | 83.3% | 3.3% | 10.0% | 3.3% |
| C | 30 | 63.3% | 26.7% | 6.7% | 3.3% |
| D | 30 | 76.7% | 3.3% | 20.0% | 0.0% |
| Dataset | 30 | 67.5% | 16.7% | 14.2% | 1.7% |

*3.4. Geometric Data Comparison between the Supervised and the Unsupervised Approach*

Figure 4 presents the comparison results between supervised and unsupervised segmentation approaches to perform tree geometric characterization from the structure of motion products. Taking into account the presence of split and merged cases in the unsupervised approach, the dataset was first analyzed not tree-by-tree but by means of the aggregation per site of each polygon identified by both segmentation methodologies. Each XY graph shows the comparison of geometric data related to four sites in both years (2017–2018). The height estimation both for tree height and crown mean height was correctly described from the proposed unsupervised method, providing $R^2$ = 1.00 correlation coefficient and a good accuracy in terms of values RMSE = 0.25 m and RMSE = 0.24 m, respectively. Considering the estimation of the crown area mean value per site, no correlation was found from the application of the proposed methods ($R^2$ = 0.01), with a high difference between the values (RMSE = 21.47 m$^2$). The crown-projected volume shows a lower correlation ($R^2$ = 0.54) than pure height-derived variables (tree height and crown mean height), but a discrete error in the absolute values (RMSE = 274.64 m$^3$).

Regarding full site characterization in terms of canopy cover area and crown projected volume, the unsupervised method provided very high correlations: $R^2 = 0.93$ and $R^2 = 0.99$, respectively.

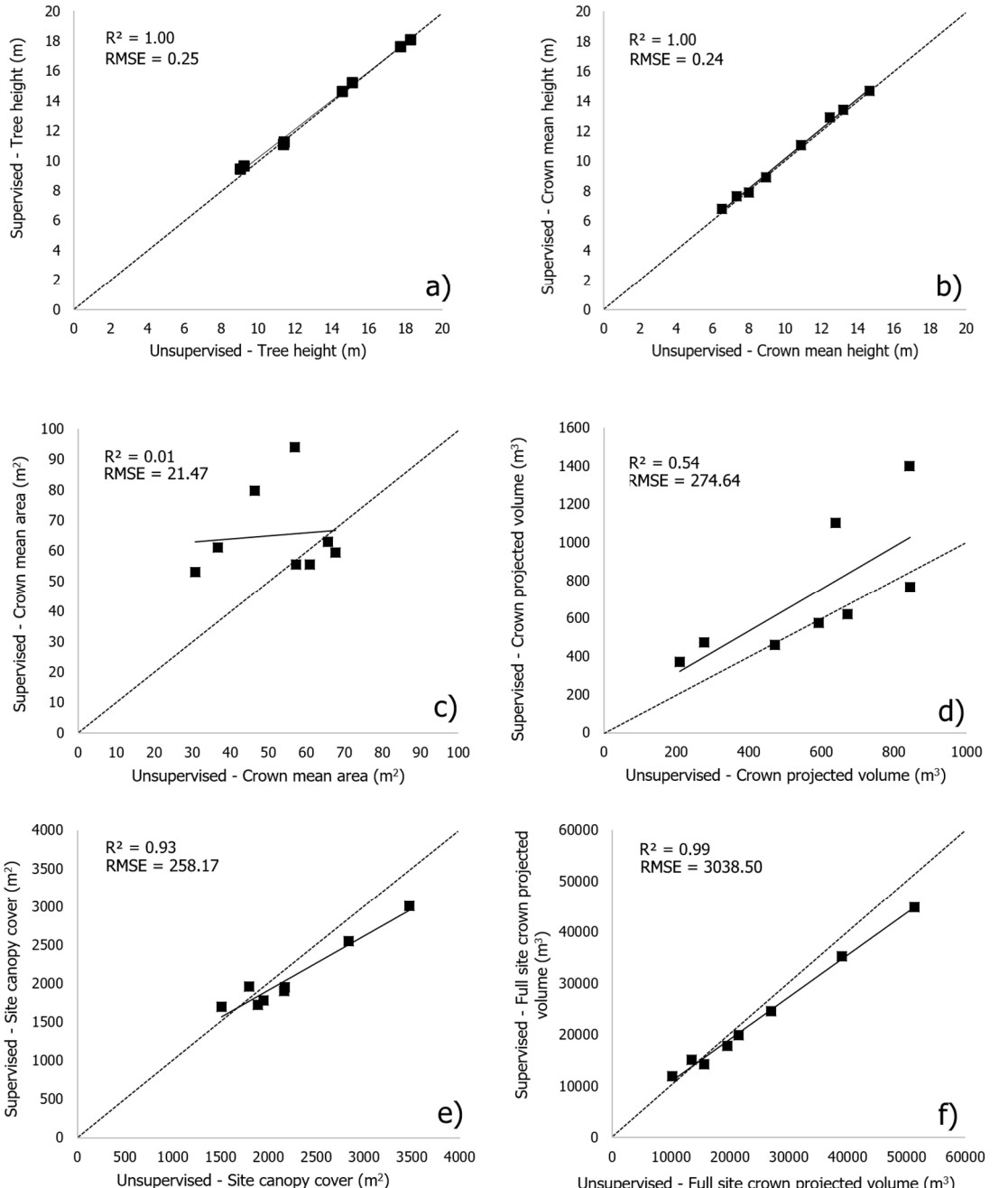

**Figure 4.** Comparison performance between supervised and unsupervised segmentation within 4 sites for 2017 and 2018 season to characterize geometric information: tree height (**a**), crown mean height (**b**), crown mean area (**c**), crown projected volume (**d**), full site canopy cover (**e**), full site crown projected volume (**f**).

A deep analysis was performed taking into account a larger dataset obtained by the unsupervised segmentation to evaluate the performance of the proposed method as a feasible tool for Wpw evaluation on large scale areas. The unsupervised dataset was created with about 67.5% matched polygons, potentially available to investigate correlation tree-to-tree with measured ground truth Wpw and supervised geometric data per tree. The dataset was then increased by adding the 16.7% of split

cases considered as single-tree data by the merging of the sub-polygons in which a sample tree was fragmented by the unsupervised approach. Figure 5 reports the correlations ($R^2$ and RSME) related to crown volume reduction of the proposed methodology versus the manually segmented mask.

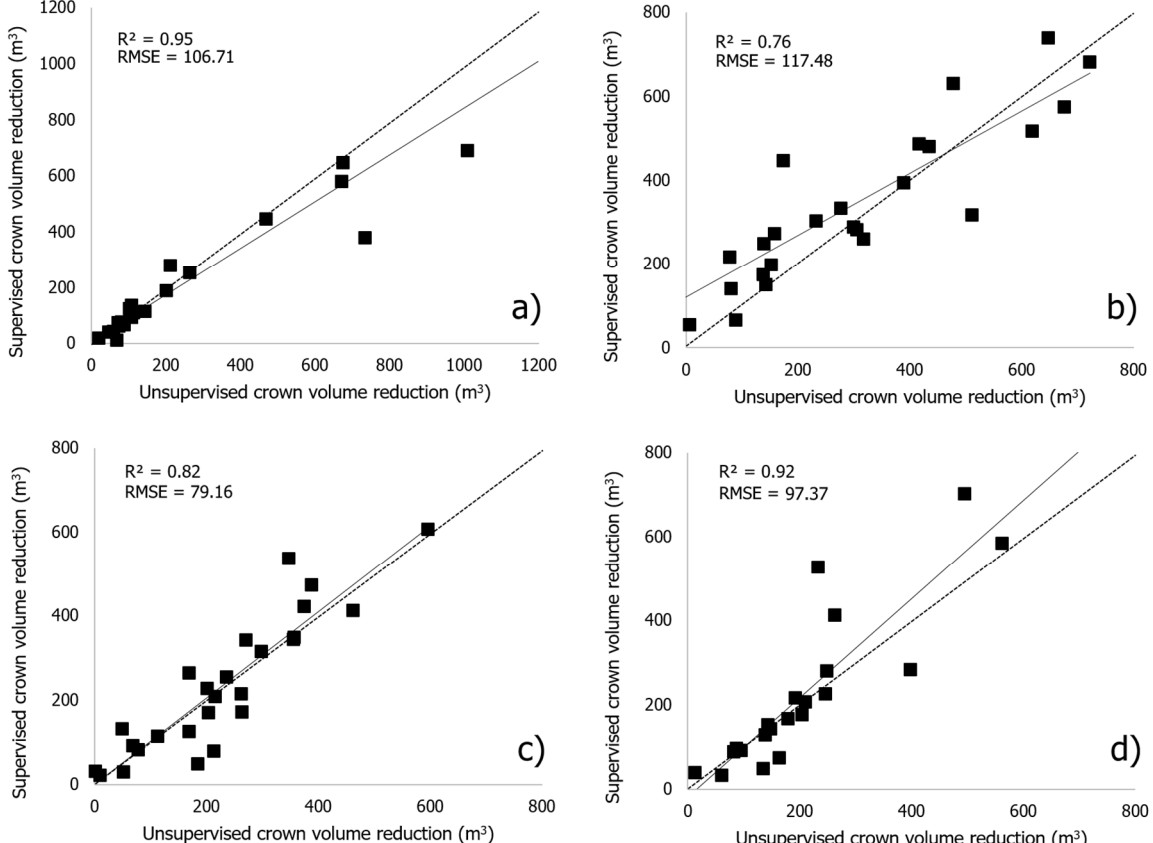

**Figure 5.** Comparison between crown projected volume reduction extracted by means of the supervised and unsupervised approach within each site (**a**, **b**, **c**, **d**). The dataset was made of both correctly segmented crown (matched) and fragmented crown (split) as a sum of each sub-polygon of the fractionated crown described in Table 4.

The unsupervised method showed a high accuracy performance in crown segmentation, providing high $R^2$ values ranging between 0.76 and 0.95, and good precision in term of absolute values, with RMSE ranging between 79.16 m$^3$ and 117.48 m$^3$. The scatterplots show results close to the 1:1 line between supervised and unsupervised segmentation methods.

*3.5. Wpw Estimation*

Table 5 presents the correlation results (equations and $R^2$) between crown projected volume reduction (X-independent variable) and pruning wood biomass (Y-dependent variable), in which a linear regression model was applied to the dataset extracted with the manually reference masks and the ground truth Wpw measurements. All sites show representative results with higher correlation coefficients for the A site ($R^2 = 0.78$), intermediate value in site C and D ($R^2 = 0.71$ and $R^2 = 0.69$ respectively) and lower in site B ($R^2 = 0.60$). Considering the similar tree ages and dimensions in the close sites C and D, Table 5 also reports the good correlations obtained by oganising the two sites as a single dataset ($R^2 = 0.65$). The linear regression analysis applied to the overall dataset provides good results but lower than the values at the single-site level ($R^2 = 0.33$).

**Table 5.** Regression analysis between ground truth Wpw data (X-independent variable) and estimated projected crown volume reduction (Y-dependent variable) obtained by the application of supervised segmentation. Linear regression results (equation and $R^2$) calculated for each site (A, B, C, D) and aggregated dataset (C + D and A + B + C + D). All liner regressions provided significance results ($p < 0.001$).

| Segmentation | Site | Equation | $R^2$ |
|---|---|---|---|
| Supervised | A | y = 1.2566x − 201.4442 | 0.78 |
| | B | y = 0.2729x + 143.1937 | 0.60 |
| | C | y = 0.3549x + 25.1030 | 0.71 |
| | D | y = 0.2028x + 64.0793 | 0.69 |
| | C + D | y = 0.2393x + 53.1303 | 0.65 |
| | A + B + C + D | y = 0.6664x + 56.446 | 0.33 |

Concerning the Wpw validation, the estimated Wpw values obtained using the calibration realized in each site with the manually segmented mask were compared with ground-truth Wpw measurements. Figure 6 shows the linear regression results within every single site and with the aggregated dataset (C + D and all dataset), which are similar to the trend found in Table 5.

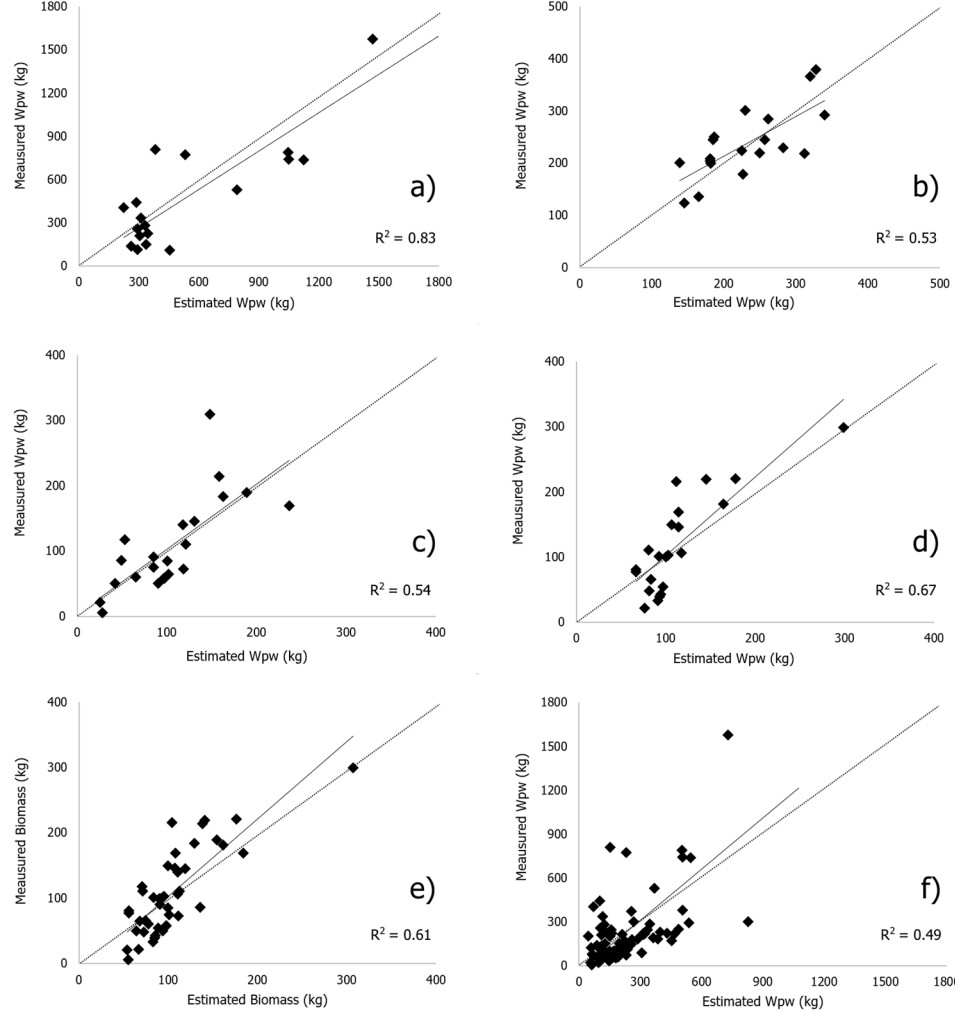

**Figure 6.** Wpw estimation validation by means of a comparison between measured and estimated Wpw retrieved by the unsupervised segmentation within site A (**a**), B (**b**), C (**c**), D (**d**), C + D (**e**) and A + B + C + D dataset (**f**).

Table 6 reports the statistic results of the methodology suggested as Wpw estimation approach. Site A presents a higher accuracy in Wpw estimation ($R^2$ = 0.83) but RMSE = 221.26 kg is very high, while site B presents lower correlations ($R^2$ = 0.53 and RMSE = 46.99). Sites C and D show good correlations with $R^2$ = 0.54 and $R^2$ = 0.67 respectively, and similar values in terms of RMSE 47.38 kg and 43.08 kg, respectively. Combining the C and D datasets, the results obtained show similar behavior to the separate dataset analysis (with $R^2$ = 0.61 and RMSE = 45.28 kg). The correlations identified using the overall dataset present a lower correlation coefficient ($R^2$ = 0.49) than the other sites, but similar to sites B and C. As reported in Table 6, the unsupervised methodology provides rRMSE values between 44.10% (site A) and 71.60% (site B), and presents very low PBias values with a minimal tendency of understimation in sites B, C, D (mean value −3.0%). Site A presents an overestimation tendency with a bias value of 12.60%.

**Table 6.** Statistic results ($R^2$, adjR2, RMSE, rRMSE (%) and PBias (%) calculated for each site (A, B, C, D) and aggregated dataset (C + D and A + B + C + D).

| Site | $R^2$ | adj$R^2$ | RMSE | rRMSE (%) | PBias (%) |
|---|---|---|---|---|---|
| A | 0.83 | 0.82 | 221.26 | 44.10 | 12.60 |
| B | 0.53 | 0.50 | 46.99 | 71.10 | −2.50 |
| C | 0.54 | 0.52 | 47.38 | 66.20 | −2.30 |
| D | 0.67 | 0.65 | 43.08 | 58.10 | −4.40 |
| C + D | 0.61 | 0.60 | 45.28 | 62.70 | −3.40 |
| A + B + C + D | 0.49 | 0.48 | 217.54 | 71.60 | −1.20 |

## 4. Discussion

This work aimed to evaluate the accuracy performance of supervised and unsupervised methodologies to estimate pruned biomass. To carry out this objective, an experimental design was planned by taking into account four sites with different conditions of vegetative growth in terms of DBH dimension, trees density and height. Site A presented the highest vegetative growth conditions, site B an intermediate level, C and D, the lowest dimensions. Those different vegetative conditions directly affected the geometric estimation provided by the UAV data analysis, so it was necessary to divide the dataset into three different groups of trees according to age and therefore, size.

Pruning intensity varied among sites as the tender's choice normally depends on the trees' growth, health conditions and age. The heavier intervention carried out with restoration purposes, as in site A, corresponded to a higher wood production, in line with analogous cases. In orchards where pruning is conducted on a long timespan (8–10 years) the amount of wood residues available for industrial purposes can be relevant (from 51.7% to 311.9% in the present study) and an early survey can provide useful information for planning the supply.

The results show that UAV monitoring has a good performance in detecting the biomass reduction after pruning, despite the differences between the trees' geometric characteristics mentioned in Section 3.1. The tree height decrease was weakly detected, mainly due to some branches not being pruned in 2017, which therefore attenuated the height reduction with their vigorous growth in 2018. Considering crown mean height and crown area, the biomass decrease is evident because they are more representative features of the whole canopy. By analyzing the data shown in Table 3, it can be stated that the crown projected volume is the best tree geometric characteristic with which biomass variation can be monitored. In site C, the highest volume decrease is not associated with the greatest height decrease and this can be explained by the typical chestnut pruning method. This technique is not characterized by a uniform topping and hedging but by the cutting of whole branches, so there is no marked height reduction. This led to the formation of crown holes whose presence can be clearly detected only by analyzing tree volume.

In the present work, the chestnut orchard condition strongly affected the segmentation accuracy. The high variability due to irregular spacing between trees, ages and dimensions, irregular crown

shape, absence of isolated tree cases, high overlap crown conditions and minimal presence of free space around each tree crown caused a lower accuracy performance in tree segmentation with respect to elevated tree detection results described in other valuable works such as those suggested by Marques et al. [76] and Jorge Torres-Sánchez et al. [90]. The large number of oversized tree crowns in the site A led to an increase in the percentage of split and merged cases, while in sites C and D, the lowest values of DBH and trees' height affect the segmentation accuracy providing 26.7% of split and 20% of merged cases respectively. Concerning the feasibility of large areas canopy cover scouting, the method proposed provided similar results as those reported by Marques et al. [76] related to an undetected tree percentage value (1.7%).

The evaluation of the accuracy of the unsupervised method applied in this study was realized by the comparison of a geometric dataset for each site in each year with the data extracted with the reference mask manually drawn. The proposed method provides a correct estimation of the mean height per tree in terms of tree height with minimal difference in absolute values (RMSE = 0.25 m), confirming the good performance as a height estimation tool for that type of survey on large areas. The mean crown area per tree within each site presents poor correlations and a high RMSE value, as a consequence of the wide value range derived from the split and merge cases of the unsupervised segmentation polygons. The crown projected volume shows a better performance in terms of correlations than the crown area due to the positive influence of the well estimated height but still with a high RMSE value. The analysis of the total canopy cover and the total crown projected volume per site provides optimal results, confirming the method as a powerful tool for fast detection in large areas. A focus elaboration on the projected crown volume reduction between the two years as a consequence of the pruning management practice was performed by increasing the dataset of the matched crowns with the sum of the sub-polygons in which some crowns were divided, reported as "split cases". The improved dataset shows the highest correlation coefficients (mean $R^2$ = 0.86) and a low difference in values (mean RMSE = 99.75 m$^3$) with respect to manually segmented crowns.

The validation of the method was carried out after a calibration step, a model was created using a regression analysis between Wpw and volume variation extracted with the reference mask manually drawn on the crown profile. Subsequently, the model identified was applied to the segmentation results obtained with the proposed method. The estimated Wpw per crown was finally correlated with the measured Wpw in order to define the accuracy in terms of correlation coefficient and RMSE. In the case of a full dataset analysis, the approach obtained good correlation ($R^2$ = 0.33 for calibration and $R^2$ = 0.49 for validation) but the clustered nature of the dataset with different tree conditions implied a lower performance, confirming the application of a site-by-site approach as the most correct choice. The method proposed showed the high coefficient of correlation ($R^2$ = 0.83) for site A, but with a very high RMSE since this site was extremely modified in terms of Wpw removed more than the others. Despite showing a lower accuracy in Wpw estimation ($R^2$ = 0.53), site B presented an acceptable RMSE = 46.99 kg. The factors that strongly reduced the segmentation success were the higher tree density within the site and an elevated overlap level between adjacent crowns, as reported in Table 3. In fact, this site is characterized by high tree and DBH values (close to the biggest in site A), but the lowest mean crown area with respect to all other sites. The application of the proposed methods in sites C and D characterized by similar tree conditions provided high and concordant performances, suggesting that they should be considered as a single dataset ($R^2$ = 0.61). This result strongly encourages the hypothesis of the feasibility of this method as a site-specific tool for large-scale monitoring. The PBias indicates an overall tendency of minimal understimation, while in site A, characterized by very different conditions in term of Wpw harvested and tree ages, the unsupervised segmentation approach shows a low overstimation of the Wpw with respect to ground truth meausurements.

In the literature, there are no studies regarding chestnut Wpw estimation using UAV. Nevertheless, there have already been recent studies that specifically derived individual biomass and $V_{SfM}$ at tree level with ITC segmentation. Among these, important references in the Mediterranean environment

were represented mainly by Guerra-Hernandez et al. [57] and Guerra-Hernandez et al. [56], who used a fixed-wing UAV equipped with an RGB camera to evaluate (i) $Wa_{SfM}$ in *Pinus pinea* regular forest plantation (10 x 16 m regular spaced, open canopy, fairly flat terrain, no understory) and (ii) $V_{SfM}$ in *Eucalyptus* regular forest plantation (3.7 × 2.5 m regularly spaced, steep terrain). Comparing our results with the aforementioned studies, it worth noting that the RMSE could only be compared with Guerra-Hernandez et al. [57], who used the same unit (kg) and reported a value of 87.46 and 117.80 kg for 2015 and 2017, respectively. These values are lower than the overall aggregated dataset RMSE (217.54 kg, Table 6) and the difference could be partially explained by the regular characteristics of the stand investigated by the reference study (spacing, tree age, field management). Thereafter, it is of pivotal importance to compare different remote sensed tree biomasses through statistic indexes that facilitate comparison between datasets or models with different scales, as rRMSE and adjR$^2$. Guerra-Hernandez et al. [57] in *P. pinea* plantation gained good results in the estimation of $Wa_{SfM}$ in comparison to measured Wa (0.85 < adjR$^2$ < 0.87 and 11.44% < rRMSE < 12.59% in two different years and model approaches) while Guerra-Hernandez et al. [56], in a *Eucalyptus* plantation, got slightly worse performances (R$^2$ = 0.43 and rRMSE = 20.31%). However, the current work presents lower correlation values (except in one case) and lower rRMSE (Table 6) with respect to the literature references, mainly due to orchard characteristics (uneven-aged and irregularly spaced) and fine pruning evaluation purposes with respect to growth monitoring. As for Guerra-Hernandez et al. [57], who focused on canopy management study in fruit production crop, our method falls within precision agriculture applications while most of the literature focused on precision forestry.

A strong point of this method was that the dataset was acquired with a low-resolution multispectral camera, which provides both geometric information from the CHM reconstruction and spectral data to calculate the NDVI layer used as a filtering approach to improve the quality of the dataset. As a consequence, the weight of the products to be processed were much lower, allowing faster data processing and requiring less computing power.

## 5. Conclusion

In the context of the Circular Economy envisaged as a "regenerative system in which resource input and waste, emission, and energy leakage are minimized by slowing, closing and narrowing material and energy loops" [92], it is important to estimate the amounts of wastes available as by-products for industrial purposes. In this specific case, chestnut pruning and their periodical availability can be forecasted and included in supply chain planning to benefit both producers and industrial users.

The unsupervised segmentation method proposed in this work made it possible to realize an accurate estimation of chestnut geometric characteristics from high-resolution CHM layers in four study sites. The results obtained are strongly in line with those extracted with a reference manually segmented mask. Applying a calibration performed on supervised UAV data extraction, the method reports a high accuracy in terms of R$^2$ and RMSE values, suggesting this approach as a fast and cost-effective tool for fast monitoring of large areas. The dataset was acquired before and after a pruning management practice in four study sites identifying three different DBH classes (around ~0.50 m, ~0.60 m, ~0.80 m). The results obtained allow for us to conclude that the method provides generally good performance, but to achieve the best Wpw estimation, is necessary to choose the correct calibration curve in the function of the DBH. This input information could be easily provided by the orchard owner, making the proposed method a useful tool for fast Wpw estimation purposes.

A future perspective could be to evaluate the potential of a combined approach analyzing also the spectral data actually used only to improve data extraction accuracy, with the aim of finding the described Wpw estimation performance by the contribution of information on vegetation indices.

**Author Contributions:** C.N. and S.F.D.G. designed the experiment. C.N., S.F.D.G., R.D., L.P., P.T., and A.M. formulated the research methodology and wrote the manuscript. A.B. performed the UAV data acquisition. All Authors reviewed and edited the draft. All authors have read and agreed to the published version of the manuscript.

**Funding:** This work was funded by Reg. UE n. 1305/2013—PSR 2014/2020—Call for "Integrated Projects of Supply chain (Progetti Integrati di Filiera—PIF) 2015". Administrative Order n. 2359 26/05/2015—P.I.F. n. 3/2015 "VACASTO PLUS". Action 16.2—Project: "OPEN RICCIO".

**Acknowledgments:** Authors' acknowledgments go to Giovanni Alessandri (Agricis Consulting s.r.l.) and Lorenzo Fazzi (Associazione Castagna del Monte Amiata) and to Niccolò Brachetti Montorselli, professional forester for the field measurements. Special thanks go to orchards' owners Mirco Fazzi and Roberto Ulivieri for the support provided during the Project.

**Conflicts of Interest:** The authors declare no conflict of interest.

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
