# Peer review of "An Automatic UAV Based Segmentation Approach for Pruning Biomass Estimation in Irregularly Spaced Chestnut Orchards"

_forests, doi:10.3390/f11030308_

Round 1
Reviewer 1 Report
This is a well-written piece of work, which goes into good detail regarding the processing and the accuracy assessment.
Some comments:
I don’t know anything about chestnut pruning, but I would suspect that the pruning takes place all around the tree crown (i.e. even the lower crown branches), and therefore the crown volume is influenced from all sides (above, side and below). The identified ‘holes’ are an interesting element worth exploring (as the authors well write in the last paragraph). Actually, when I read about Tetracam, I expected some spectral analysis to support the volume measurements.
Some minor corrections:
Line 152. How were the sample trees selected? If random, please specify “randomly selected”.
Lines 179-182. What ground truth data was loaded in Agisoft Metashape software for geometrically correcting the UAV mosaic, and for assessing the point cloud accuracy?
Line 187. Why 0.25m? What was the density of the point cloud in terms of points per square metre? Does the density justify the 0.25m?
Line 248. Best to have the title “Results” in the next page.
Table 5. What’s the point of all the Signif.codes when only one of them is displayed in the table? I suggest removal, and adding a small note in the Table 5 caption.
Line 373. “…trees density and height.”
Line 376. “…divided into three different groups…”
Line 461 and line 466 “…in four study sites…”
Author Response
This is a well-written piece of work, which goes into good detail regarding the processing and the accuracy assessment.
Some comments:
I don’t know anything about chestnut pruning, but I would suspect that the pruning takes place all around the tree crown (i.e. even the lower crown branches), and therefore the crown volume is influenced from all sides (above, side and below). The identified ‘holes’ are an interesting element worth exploring (as the authors well write in the last paragraph). Actually, when I read about Tetracam, I expected some spectral analysis to support the volume measurements.
We would like to thank the reviewer for the precious suggestions provided in this review.
Taking your last comment into account, instead we exploited the multispectral dataset as a support layer (filter) to remove the small holes within each canopy before to perform tree volume estimation.
The dataset was reprocessed to calculate the additional statistics requested by a reviewer which were summarized in table 6. Some errors were identified in the dataset used to make table 5, consequently the data in figure 6 were also modified. However, the trends and R2 have remained similar. The authors apologize for the mistake.
Some minor corrections:
Line 152. How were the sample trees selected? If random, please specify “randomly selected”.
The choice of the sample trees was made by identifying representative plants for each site in terms of size (DBH). Sentence added in section 2.2
Lines 179-182. What ground truth data was loaded in Agisoft Metashape software for geometrically correcting the UAV mosaic, and for assessing the point cloud accuracy?
The reviewer highlights an important factor for carrying out high precision photogrammetric analysis. Unfortunately, no ground control points were set up during the flight campaigns on the sites. However, according to the authors' experience, the reduced size of the sites (0.3-0.5 ha) and the high overlap of the acquired images, they made it possible to obtain good products for this type of analysis.
Line 187. Why 0.25m? What was the density of the point cloud in terms of points per square metre? Does the density justify the 0.25m?
The reviewer is right, the description of the method was not clear. Accordingly, the text has been modified.
Line 248. Best to have the title “Results” in the next page.
Done
Table 5. What’s the point of all the Signif.codes when only one of them is displayed in the table? I suggest removal, and adding a small note in the Table 5 caption.
Done
Line 373. “…trees density and height.”
Done
Line 376. “…divided into three different groups…”
Done
Line 461 and line 466 “…in four study sites…”
Done

Reviewer 2 Report
Reviewer
General comments:
This paper examines the use of Multiespectral-UAV imagery for the estimation of pruning biomass through differences in the volume of canopy trees and evaluate the performance of an unsupervised segmentation methodology as a feasible tool for large areas analysis on uneven-aged and irregularly spaced chestnut (Castanea sativa Mill.) orchards. As the author mentioned, there are some works about monitoring forest biomass using imagery data from UAV. The literature also seems to have a lack of studies at tree level in which pruning residues coming from canopy management were evaluated comprehensively on DAP((Digital Aerial Photogrammetry) -SfM-based products. This is a good systematic analysis for 3D-modeling to determine biomass estimation at tree level. The introduction is comprehensive and well written. However, the method and results must be improved in the manuscript. Methodologically the paper has some faults, since the analysis is not thorough due to the lack of presentation of results comparing with ground truth field data and the lack of a more detailed description of the modelling stage at method and result section. The full scope of the types of cover and structural conditions where SfM-derived products that can be used as an alternative to ALS (Airbone Laser Scanning) data to estimate biomass at tree level also continues to be an active area of research. The manuscript can be considered a valuable contribution to the literature and should be considered for publishing. However, the following corrections should be considered and done. It important clarify along the manuscript that you are modelling only pruning biomass from SfM-based products. Abbreviation of the term “pruning wood biomass” must be included along the manuscript.
Specific Comments
Introduction
Introduction does not provide sufficient background and not include all relevant references using the same technologies for 3D-modeling of biomass in other forest type using multiepectral imagery or DAP (Digital Aerial Photogrammetry)-points clouds from UAV. This relevant references must be also cited and included along the manuscript, especially in the introduction and discussion. The authors should discuss the results and how they can be interpreted in perspective of these previous studies in the discussion. The findings and their implications should be discussed in the broadest context possible.
For example, I think it is important to hightlight in the introduction that there are already recent studies that specifically derived individual biomass and volume at tree level and compare your models in terms of adjR2 and rRMSE(%) performance with this studies. For example, in the line 82 you mentioned some studies as extrapolation of further dendrometric parameters (i.e. AGB) [55–57], but any of this studies you cited are example of estimating aboveground biomass using canopy segmentation from the canopy height model (CHM) from UAV imagery at tree level.
Please include specific studies to estimate biomass and described briefly that two main strategies have been adopted for DAP and ALS-based analysis in forestry inventories:(i) the Area-Based Approach (ABA), a distribution-based technique which typically provides data at stand level, and (ii) the individual tree crown segmentation (ITC) delineation, in which individual tree crowns, heights and positions are the basic units of assessment. You have to mention specific studies to estimative biomass at stand level (i. e Kachamba et al. 2017 , Li et al 2019, Domingo et al., 2019) and at tree level as your study (i.e Guerra-Hernandez et al., 2017, 2019, in open Mediterranean forest of coniferous Pinus pinea (Central of Portugal) using DAP point clouds from UAV and Evergreen Eucalyptus spp. (North of Portugal)). The last ones are important references of the methodology the authors could be tested and followed to modeling SfM individual tree diameter and SfM-derived individual tree aboveground biomass and volume from the CHM in different forest types.
Please according IUFRO (International Union of Forest Research Organization) terminology, aboveground biomass must be abbreviated as Wa. If you refer to field data biomass, you can abbreviate as Wa. If you are using aboveground biomass estimation from Remote sensing (RS) data you can use WaSfM for example.
Please if you are using only pruning wood biomass you can use something abbreviation as Wpw. It is not clear to the reader what component from biomass are you modeling.
References
Domingo, D., Ørka, H. O., Næsset, E., Kachamba, D., & Gobakken, T. (2019). Effects of uav image resolution, camera type, and image overlap on accuracy of biomass predictions in a tropical woodland. Remote Sensing, 11(8), 948.
Guerra-Hernández, Juan, Eduardo González-Ferreiro, Vicente J. Monleón, Sonia P. Faias, Margarida Tomé, and Ramón A. Díaz-Varela. 2017. ‘Use of Multi-Temporal UAV-Derived Imagery for Estimating Individual Tree Growth in Pinus Pinea Stands’. Forests 8 (8):300.
Kachamba, Daud Jones, Hans Ole Ørka, Erik Næsset, Tron Eid, and Terje Gobakken. 2017. ‘Influence of Plot Size on Efficiency of Biomass Estimates in Inventories of Dry Tropical Forests Assisted by Photogrammetric Data from an Unmanned Aircraft System’. Remote Sensing 9 (6):610.
Guerra-Hernández, J.; Cosenza, D.N.; Cardil, A.; Silva, C.A.; Botequim, B.; Soares, P.; Silva, M.; González-Ferreiro, E.; Díaz-Varela, R.A. Predicting Growing Stock Volume of Eucalyptus Plantations Using 3-D Point Clouds Derived from UAV Imagery and ALS Data. Forests 2019, 10, 905.
Li, Z.; Zan, Q.; Yang, Q.; Zhu, D.; Chen, Y.; Yu, S. Remote estimation of mangrove aboveground carbon 634 stock at the species level using a low-cost unmanned aerial vehicle system. Remote Sens. 2019, 11(9), 635 1018. 636
Material and Method
2.1. Experimental site
Table 1. Please if it is possible you can include approximately the canopy cover (%) by study site from field data. You indicated the density but if you have available summer LiDAR data from the study areas you can estimate CC% also from this RS source. This variable is critical in dense canopy cover (CC) and steep rugged terrain to estimate accurate tree height variables from the CHM. Please include the number of trees you used to calculate pruning biomass by study site
2.3. UAV platform and data processing
L167 The authors do not indicate error in the geolocation accuracy of ground control points (GCP) in the study areas during SfM reconstruction using Agisoft Professional Edition 1.5.2. The authors need to indicate clearly in the methods and results what happened and what was done, and then make sure in the discussion that conclusions drawn from the results regarding SfM-DEM is clearly framed in the understanding that the results could be affected by a poor accurate topographic survey in the mark of CGPs, resulting in wrong orientation and scale of the reconstruction of your 3D model in forest areas. For reliable accuracy of GPS measurement, all GCPs must be located in open areas with no canopy cover but I understand that sometimes this is not possible in dense and continuous canopy cover.
You can read some technical report and articles (Westoby et al. 2012, Mesas-Carrascosa et al. 2015; Agüera-Vega, Carvajal-Ramírez, and Martínez-Carricondo 2016; Mesas-Carrascosa et al. 2016; Tomaštík et al. 2017, Guerra Hernandez et al., 2017, 2018) on the problem of not using control points in a precise way, or without using GCPs (Gašparović et al. 2017) . We already know this approach simplifies the unambiguous co-location of image and object space targets and also ensures a reliable, well-distributed network of targets across the area of interest, enabling an assessment of any non-linear structural errors in the 3D-SfM reconstruction. https://support.pix4d.com/hc/en-us/articles/202558909#gsc.tab=0. A GCP error superior to 2 times the Ground Sampling Distance (GSD) may indicate a severe issue with the dataset or more likely an error when marking or specifying the GCPs. You also could indicate the accuracy of your GPS using the GNSS-Real-Time-Kinematic (RTK) method, specifying clearly in the text the technical data of the Horizontal and Vertical Accuracy assuming normal favorable conditions. (e.g Horizontal: 8 mm + 1 ppm (rms) Vertical: 15 mm + 1 ppm (rms)) due to the importance of GCPs for image orientation and DSM generation.
If you did no use CGPs, please indicate clearly you do not use CGP during SfM process to generate the point cloud
L185 Please indicate clearly how do you normalize the height from the point cloud to obtain de canopy height model (CHM) from Lastools. Did you use a digital elevation model (DEM) from automatic classification of ground points from Photogrammetric software or did you use other available LiDAR data from this area to obtain the DEM in order to normalize your data? This question is critical in dense canopy cover (CC) and steep rugged terrain to estimate accurate tree height variables from the CHM (see Guerra-Hernández et al., 2018).
References
Agüera-Vega, Francisco, Fernando Carvajal-Ramírez, and Patricio Martínez-Carricondo. 2016. ‘Accuracy of Digital Surface Models and Orthophotos Derived from Unmanned Aerial Vehicle Photogrammetry’. Journal of Surveying Engineering, 04016025.
Chen, Shijuan, Gregory J. McDermid, Guillermo Castilla, and Julia Linke. 2017. ‘Measuring Vegetation Height in Linear Disturbances in the Boreal Forest with UAV Photogrammetry’. Remote Sensing 9 (12):1257.
Birdal, A. C., Avdan, U., & Türk, T. (2017). Estimating tree heights with images from an unmanned aerial vehicle. Geomatics, Natural Hazards and Risk, 8(2), 1144-1156.
Goodbody, Tristan RH, Nicholas C. Coops, Txomin Hermosilla, Piotr Tompalski, and Patrick Crawford. 2017. ‘Assessing the Status of Forest Regeneration Using Digital Aerial Photogrammetry and Unmanned Aerial Systems’. International Journal of Remote Sensing, 1–19.
Goodbody, Tristan RH, Nicholas C. Coops, Peter L. Marshall, Piotr Tompalski, and Patrick Crawford. 2017. ‘Unmanned Aerial Systems for Precision Forest Inventory Purposes: A Review and Case Study’. The Forestry Chronicle 93 (1):71–81.
Gašparović, M., Seletković, A., Berta, A., & Balenović, I. (2017). The evaluation of photogrammetry-based DSM from low-cost UAV by LiDAR-based DSM. South-east European forestry, 8(2), 117-125.
Guerra-Hernández, Juan, Eduardo González-Ferreiro, Vicente J. Monleón, Sonia P. Faias, Margarida Tomé, and Ramón A. Díaz-Varela. 2017. ‘Use of Multi-Temporal UAV-Derived Imagery for Estimating Individual Tree Growth in Pinus Pinea Stands’. Forests 8 (8):300.
Guerra-Hernández, J., Cosenza, D. N., Rodriguez, L. C. E., Silva, M., Tomé, M., Díaz-Varela, R. A., & González-Ferreiro, E. (2018). Comparison of ALS-and UAV (SfM)-derived high-density point clouds for individual tree detection in Eucalyptus plantations. International Journal of Remote Sensing, 39(15-16), 5211-5235.
Jensen, Jennifer L R, and Adam J. Mathews. 2016. ‘Assessment of Image-Based Point Cloud Products to Generate a Bare Earth Surface and Estimate Canopy Heights in a Woodland Ecosystem’. Remote Sensing 8 (1). https://doi.org/10.3390/rs8010050.
Mesas-Carrascosa, Francisco-Javier, María Dolores Notario García, Jose Emilio Meroño de Larriva, and Alfonso García-Ferrer. 2016. ‘An Analysis of the Influence of Flight Parameters in the Generation of Unmanned Aerial Vehicle (UAV) Orthomosaicks to Survey Archaeological Areas’. Sensors 16 (11):1838.
Mesas-Carrascosa, Francisco-Javier, Jorge Torres-Sánchez, Inmaculada Clavero-Rumbao, Alfonso García-Ferrer, Jose-Manuel Peña, Irene Borra-Serrano, and Francisca López-Granados. 2015. ‘Assessing Optimal Flight Parameters for Generating Accurate Multispectral
Orthomosaicks by UAV to Support Site-Specific Crop Management’. Remote Sensing 7 (10):12793–814.
Tomaštík, J., Mokroš, M., Saloň, Š., Chudý, F., & Tunák, D. (2017). Accuracy of photogrammetric UAV-based point clouds under conditions of partially-open forest canopy. Forests, 8(5), 151.
Line 151 2.2. Biomass ground measurement. Please specify are modelling only pruning biomass “2.2. Prunnig Biomass ground measurement”.
Line 239 2.5 Biomass estimation
Please describe the linear equation and variables you used to modelling the pruning biomass using remote sensing data
where Y is the dependent variable, X1 (crown projected volume reduction) are independent variables β0 is the intercept, β1 is the parameters to be estimated by linear regression analysis and ε is the additive term of the error. The models were fitted using the lm function implemented in the BASE package of of R software (R Core Team, 2019).
The coefficient of determination (adjR2, Eq 3), the overall root mean square error (RMSE, Eq. 4), the relative root mean square error (rRMSE, Eq 5) and the bias (Eq. 6) to determine the accuracy of SfM models for estimating biomass using crown projected volume reduction must be included in material and method.
????2=1−((?−1)Σ(????=1−?̂?)2(?−?)Σ(????=1−?̅?)2)
(3)
RMSE=√Σ(????=1−ŷ?)2?
(4)
rRMSE=?????̅
(5)
Bias=Σ(?̂?− ????=1)?
(6)
where n is the number of trees, yi is the field measured tree biomass i, ?̅ is the the mean observed value of biomass, ?̂? is the estimated value of biomass derived from the linear regression model and ? is the number of parameters from de the models.
It is important note that two approaches are possible when you are estimating individual tree biomass or volume (see 2.6 in Guerra-Hernandez et al., 2019 and or Alonso et al 2020). In the first approach, a model is fitted by using field data dbh as the dependent variable and hSfM, caSfM, hALS and caALS variables from the segmentation were the explanatory variables. Then, the predicted diameter and hSfM and hALS could be included as explanatory variables in Eq .1 to predict biomass (WaSfM) for the subset of trees for SfM. A second approach,a model in Eq. 2 was fitted to predict WaSfM for the trees, respectively but Wa (calculated using the field-measured d and h in Eq. 1) was used as a dependent variable, and hSfM, caSfM , hALS and caALS were the explanatory variables. In your case, I understood that explanatory variables of your model is crown projected volume. Please use abbreviate the variables you use in your model and use italic if it is possible.
Result (1)
+++++=nnXβXβXββY22110
Line 302 3.4. Geometric data comparison between supervised and unsupervised approach
Finally It important clarify along the manuscript that you are not using field data as ground truth data to estimate tree height, crown area, and crown projected volume for each tree and only are comparing RS products as the figure 4 and figure 5
Please include 1:1 line in all your Scatterplots (Figure 4, 5) comparing RS- derived variables against field-derived variables. It is the only way to know when you are modelling a variable if your model are underestimating or overestimating respect to your ground truth data.
Line 340 3.5. Biomass estimation
Table 5. Please include the abbreviation variable you are using into the model. I think it is the crown projected volume reduction but I am not sure?
Although you have the values in the figure 6 I think it is better include in this tables also your values of the overall root mean square error (RMSE, Eq. 4), the relative root mean square error (rRMSE, Eq 5) to determine the accuracy of the models for estimating biomass using crown projected volume reduction
Figure 6. Please include 1:1 line in all your Scatterplots (Figure 6) comparing RS- derived biolmass against field-derived biomass data. It is the only way to know when you are modelling a variable if your model are underestimating or overestimating respect to your field-derived biomass data.
In the case your linear regression estimation have a bias, underestimating or overestimating respect to your ground truth biomass data you. Please you have to disscuss the possible reason of this bias of you model
Figure 6. Please include abbreviation variables for pruning wood biomass measured and estimated pruning wood biomass and in all your axis x and y label of the Scatterplots (Figure 6)
Discussion
If the article is published, I think that the discussion section must also include a small paragraph where adjR2 ,RMSE and rRMSE (%) of your models are discussed with the values obtained by other authors in the literature from other forest species or plantation modelling biomass at tree level. I think it is important calculate and include also de rRMSE of your models since it is easier to compare with other values in the literature
For example, regarding biomass modelling, in the site A, the performance of the pruning wood biomass estimates model for predicting tree biomass directly from SfM-derived variables (R2=0.78) are slightly lower to a Guerra-Hernandez et at., 2017 models in terms of R2 for open P. pinea plantation (see table 4, Mef or R2= 0.84 in 2015, Mef or R2=0.85 in 2017) modelling aboveground biomass directly from SfM variables from the image (tree height and crown area) using 52 trees.
Once you include 1:1 line in all your Scatterplots (Figure 6) will be possible to discuss a possible bias of your models estimation throughout the observed biomass range from field data.
L 441-444 You can not use this references to compare your results from your study since the results of these studies are at stand level in tropical forest and you results you presents are at tree level
References
Guerra-Hernández, Juan, Eduardo González-Ferreiro, Vicente J. Monleón, Sonia P. Faias, Margarida Tomé, and Ramón A. Díaz-Varela. 2017. ‘Use of Multi-Temporal UAV-Derived Imagery for Estimating Individual Tree Growth in Pinus Pinea Stands’. Forests 8 (8):300.
Author Response
The authors would like to thank the reviewer for all of his/her careful, constructive and insightful comments in relation to this work. We outline in the attached file our responses to these comments, and where appropriate indicate what has been changed in the manuscript as a result. For ease of viewing, the authors responses to reviewer comments are shown in red (Please see the attachment). We believe that these comments help greatly in making our manuscript clearer and more useful to the reader.

Round 2
Reviewer 2 Report
Minor comments
L213 You have to include "normalize points cloud to obtain the CHM was done using use a digital elevation model (DEM) from automatic classification of ground points from Photogrammetric software"
Author Response
Hi, your remark at Line 213 has been added to the manuscript
